# Consistent Direct Diffusion Bridge with Injection for MRI Reconstruction

**Sergey Pyatkovskiy**[1]                                  PYATKOVSKIY@UNIST.AC.KR

**Jaejun Yoo**[1,†]                                        JAEJUN.YOO@UNIST.AC.KR

[1] *50, UNIST-gil, Eonyang-eup, Ulju-gun, Ulsan, Graduate School of Artificial Intelligence (AIGS), Ulsan National Institute of Science and Technology (UNIST), South Korea.*

## Abstract

In MRI, lengthy acquisition times often cause motion artifacts. We propose a new method, Consistent Direct Diffusion Bridge with Injection (CCDBI), which leverages diffusion-based image priors to reconstruct MRI from undersampled $k$-space data. Unlike traditional diffusion methods starting with Gaussian noise, CCDBI begins sampling from actual measurements, improving accuracy by aligning with the target image. By combining information from noisy image domain and $k$-space adaptively, CCDBI ensures consistency in the Direct Diffusion Bridge (DDB), enhancing reconstruction quality. Experimental results on IXI and OASIS-2 datasets demonstrate CCDBI's superiority over existing algorithms.

**Keywords:** Magnetic Resonance Imaging, Diffusion Bridge, Reconstruction.

## 1. Introduction

MRI's lengthy data acquisition poses challenges: increased motion artifacts and limited patient access, potentially delaying diagnosis. Recent studies have suggested using diffusion-based image prior to enhance performance for imagining tasks (Chung et al., 2022). However, diffusion models face challenges in inverse problem tasks due to its slow reconstruction and hallucination, arising from the disparity between the Gaussian prior distribution and actual data distribution.

The Direct Diffusion Bridge (DDB) (Chung et al., 2023), a novel adaptation of standard diffusion models, starts directly from measurements, providing a more accurate starting point aligned with the target image. While this reduces required sampling steps significantly, DDB still lacks data sampling consistency. To address this, CDDB (Chung et al., 2023) has applied standard diffusion guidance yet the authors have not thoroughly assessed its impact on DDB's sampling process.

In this paper, we investigate the effective conditioning methods for DDB sampling algorithms that utilize $k$-space information, aiming to avoid any potential adverse effects on the model's performance. We refer to this new method as Consistent Direct Diffusion Bridge with Injection (CCDBI), Algorithm 3 and Figure 1. Our experiments on IXI and OASIS-2 datasets show that CCDBI outperforms existing algorithms.

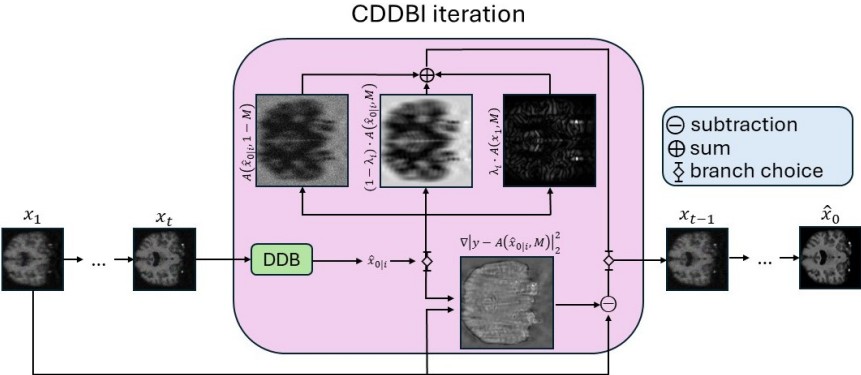

Figure 1: Overview of the proposed method. CCDBI utilized the pretrained DDB for iterative generation starting from $x_1$ and generating $\hat{x}_0$ applying the proposed approach per each iteration choosing branch to apply following chosen scheme.

---

**Algorithm 1:** CDDBI ($\boldsymbol{thres}_{big} = 0.95$, $\boldsymbol{thres}_{small_{IXI/OASIS}} = 0.7/0.1$)

---

**Input:** $\boldsymbol{\alpha}, \boldsymbol{\phi}, N, \boldsymbol{y}, \boldsymbol{M}, \boldsymbol{thres}_{big/small} \in (0,1), \boldsymbol{\xi}, \boldsymbol{\lambda}, d_0 = d_1 = 1$

**Output:** $\hat{x}_0$

**for** $i = N - 1$ *to* $0$ **do**

    $1 : \hat{\boldsymbol{x}}_{\boldsymbol{0}|\boldsymbol{i}} \leftarrow \boldsymbol{x}_{\boldsymbol{i}} + \phi_i \cdot s_\theta(\boldsymbol{x}_{\boldsymbol{i}}, i); \boldsymbol{z} \sim \mathcal{N}(\boldsymbol{0}, \boldsymbol{I}); r \sim \mathcal{U}(0, 1);$ // $\hat{x}_0$ prediction at timestep $i$

    $2 : \boldsymbol{thres} = \begin{cases} \boldsymbol{thres}_{big} & d_0 < d_1 \\ \boldsymbol{thres}_{small} & \text{otherwise} \end{cases}$     // Adaptive threshold based on $d_0$ and $d_1$

    $3$ :**if** $\boldsymbol{thres} > r$ **then**

        $4 : \hat{\boldsymbol{x}}_{\boldsymbol{0}|\boldsymbol{i}} \leftarrow \mathcal{F}^{-1}((1 - \boldsymbol{M})\mathcal{F}(\hat{\boldsymbol{x}}_{\boldsymbol{0}}) + \lambda_i \boldsymbol{y} + (1 - \lambda_i)\boldsymbol{M}\mathcal{F}(\hat{\boldsymbol{x}}_{\boldsymbol{0}}))$     // $\boldsymbol{y}$ injection

    **end**

    $5 : \boldsymbol{x}_{\boldsymbol{i-1}} \leftarrow (1 - \alpha_{i-1|i}^2)\hat{\boldsymbol{x}}_{\boldsymbol{0}|\boldsymbol{i}} + \alpha_{i-1|i}^2 \cdot \boldsymbol{x}_{\boldsymbol{i}} + \sigma_{i-1|i} \cdot \boldsymbol{z}$

    $6$ :**if** $\boldsymbol{thres} \leq r$ **then**

        $7 : \boldsymbol{x}_{\boldsymbol{i-1}} \leftarrow \boldsymbol{x}_{\boldsymbol{i-1}} - \xi_i \cdot \nabla_{\boldsymbol{x}_{\boldsymbol{i}}} \|\boldsymbol{y} - \boldsymbol{M} \cdot \mathcal{F}(\hat{\boldsymbol{x}}_{\boldsymbol{0}|\boldsymbol{i}})\|_2^2$     // DPS guidance

        $8 : d_1 \leftarrow d_0; \quad d_0 \leftarrow \|\boldsymbol{y} - \boldsymbol{M} \cdot \mathcal{F}(\hat{\boldsymbol{x}}_{\boldsymbol{0}|\boldsymbol{i}})\|_2^2$     // Storing difference norms

    **end**

**end**

---

## 2. Method

Following the approach outlined in (Chung et al., 2023), the inverse sampling algorithm integrates DPS (Chung et al., 2022) consistency-imposing gradient steps between reverse diffusion steps. A guidance term (Algoritm 3, lines 7,8) is introduced to align current timestep predictions with the provided measurements. The guidance via $\nabla_{x_i} \|y - M \cdot \mathcal{F}(\hat{x}_0)\|_2^2$ for unconditional model is necessary to enforce conditional sampling.

However, DPS guidance could potentially have a negative impact on DDB, as it incorporates prior (measurement) information during the training process. To prevent potential perturbations, we propose to constrain the usage of DPS. Specifically, in Algoritm 3 (line 6)

---

**Algorithm 2:** CDDBI 2 no time separated

---

**Input:** $\boldsymbol{\alpha}, \boldsymbol{\phi}, N, \boldsymbol{y}, \boldsymbol{M}, \boldsymbol{thres}_{default} \in (0,1), \boldsymbol{\xi}, d_0 = d_1 = 1$

**Output:** $\hat{x}_0$

**for** $i = N - 1$ *to* 0 **do**

    $1 : \hat{\boldsymbol{x}}_{\boldsymbol{0}|\boldsymbol{i}} \leftarrow \boldsymbol{x}_{\boldsymbol{i}} + \phi_i \cdot s_\theta(\boldsymbol{x}_{\boldsymbol{i}}, i); \boldsymbol{z} \sim \mathcal{N}(\boldsymbol{0}, \boldsymbol{I}); r \sim \mathcal{U}(0, 1);$ // $\hat{x}_0$ prediction at timestep $i$

    $2 : \boldsymbol{thres} \leftarrow \boldsymbol{thres}_{default} * \frac{d_0}{d_1}$               // thres=clamp(thres, 0.05, 0.95)

    $3 : \lambda_i \leftarrow \boldsymbol{thres} * sin(...)$

    $4 : \hat{\boldsymbol{x}}_{\boldsymbol{0}|\boldsymbol{i}} \leftarrow ((1 - \boldsymbol{M})\hat{\boldsymbol{x}}_{\boldsymbol{0}} + \lambda_i \boldsymbol{y} + (1 - \lambda_i)\boldsymbol{M}\hat{\boldsymbol{x}}_{\boldsymbol{0}})$          // $\boldsymbol{y}$ injection

    $5 : \boldsymbol{x}_{\boldsymbol{i-1}} \leftarrow (1 - \alpha_{i-1|i}^2)\hat{\boldsymbol{x}}_{\boldsymbol{0}|\boldsymbol{i}} + \alpha_{i-1|i}^2 \cdot \boldsymbol{x}_{\boldsymbol{i}} + \sigma_{i-1|i} \cdot \boldsymbol{z}$

    $6 : \xi_i \leftarrow \xi_{max} * (1 - \boldsymbol{thres}) * sin(...)$

    $7 : \boldsymbol{x}_{\boldsymbol{i-1}} \leftarrow \boldsymbol{x}_{\boldsymbol{i-1}} - \xi_i \cdot \nabla_{\boldsymbol{x}_{\boldsymbol{i}}} \|\boldsymbol{y} - \boldsymbol{M} \cdot (\hat{\boldsymbol{x}}_{\boldsymbol{0}|\boldsymbol{i}})\|_2^2$          // DPS guidance

    $8 : d_1 \leftarrow d_0; \quad d_0 \leftarrow \|\boldsymbol{y} - \boldsymbol{M} \cdot (\hat{\boldsymbol{x}}_{\boldsymbol{0}|\boldsymbol{i}})\|_2^2$          // Storing difference norms

**end**

---

---

**Algorithm 3:** CDDBI 3 no mask

---

**Input:** $\boldsymbol{\alpha}, \boldsymbol{\phi}, N, \boldsymbol{y}, \boldsymbol{M}, \boldsymbol{thres}_{big/small} \in (0,1), \boldsymbol{\xi}, \boldsymbol{\lambda}, d_0 = d_1 = 1$

**Output:** $\hat{x}_0$

**for** $i = N - 1$ *to* 0 **do**

    $1 : \hat{\boldsymbol{x}}_{\boldsymbol{0}|\boldsymbol{i}} \leftarrow \boldsymbol{x}_{\boldsymbol{i}} + \phi_i \cdot s_\theta(\boldsymbol{x}_{\boldsymbol{i}}, i); \boldsymbol{z} \sim \mathcal{N}(\boldsymbol{0}, \boldsymbol{I}); r \sim \mathcal{U}(0, 1);$ // $\hat{x}_0$ prediction at timestep $i$

    $2 : \boldsymbol{thres} \leftarrow \boldsymbol{thres} + 1 - \frac{d_0}{d_1}$             // thres=clamp(thres, 0.05, 0.95)

    $3 : \lambda_i \leftarrow \lambda_{max} * \boldsymbol{thres} * sin(...)$

    $4 : \hat{\boldsymbol{x}}_{\boldsymbol{0}|\boldsymbol{i}} \leftarrow (\lambda_i \boldsymbol{y} + (1 - \lambda_i)\hat{\boldsymbol{x}}_{\boldsymbol{0}})$          // $\boldsymbol{y}$ injection

    $5 : \boldsymbol{x}_{\boldsymbol{i-1}} \leftarrow (1 - \alpha_{i-1|i}^2)\hat{\boldsymbol{x}}_{\boldsymbol{0}|\boldsymbol{i}} + \alpha_{i-1|i}^2 \cdot \boldsymbol{x}_{\boldsymbol{i}} + \sigma_{i-1|i} \cdot \boldsymbol{z}$

    $6 : \xi_i \leftarrow \xi_{max} * (1 - \boldsymbol{thres}) * sin(...)$

    $7 : \boldsymbol{x}_{\boldsymbol{i-1}} \leftarrow \boldsymbol{x}_{\boldsymbol{i-1}} - \xi_i \cdot \nabla_{\boldsymbol{x}_{\boldsymbol{i}}} \|\boldsymbol{y} - A \cdot (\hat{\boldsymbol{x}}_{\boldsymbol{0}|\boldsymbol{i}})\|_2^2$          // DPS guidance

    $8 : d_1 \leftarrow d_0; \quad d_0 \leftarrow \|\boldsymbol{y} - A \cdot (\hat{\boldsymbol{x}}_{\boldsymbol{0}|\boldsymbol{i}})\|_2^2$          // Storing difference norms

**end**

---

`thres` limits DPS activity during the sampling process. To balance the constraining effect and enhance DDB's generation consistency, we propose to introduce a conditional mechanism over known regions (mask $M$) in $k$-space, which is referred to as Consistent Direct Diffusion Bridge with Injection (CCDBI). Our method substitute the network reconstruction of initially known regions with the original measurements (injection) via a weighted sum between the measurement and the timestep $i$th masked prediction (line 4 in Algorithm 3). Note that operating only over known areas of measurement $\boldsymbol{y}$ provides local conditioning effect. The hyperparameter $\lambda$ is a scale that determines the influence of measurement over final prediction.

Given the contrasting characteristics of the injection and DPS parts in the sampling process, we further propose to switch between two modes (Algorithm 3, lines 3,6) by introducing an adaptive threshold, denoted as **thres**. Specifically, we compare the current and previous values of $\|\boldsymbol{y} - M \cdot \mathcal{F}(\hat{\boldsymbol{x}}_{\boldsymbol{0}|\boldsymbol{i}})\|_2^2$ and check whether the algorithm starts priori-

Table 1: Comparison results on IXI and OASIS-2 datasets.

| | RefGan | FDB | DPS | I2SB | CDDB | Ours (w/o Injection) | Ours (w/o DPS) | Ours (full) |
|---|---|---|---|---|---|---|---|---|
| **IXI, Mask x4** | | | | | | | | |
| PSNR ↑ | 21.78 | 14.62 | 21.13 | 22.25 | 22.27 | 22.30 | 22.40 | **22.72** |
| SSIM ↑ | 0.42 | 0.48 | 0.58 | 0.67 | 0.67 | 0.67 | 0.68 | **0.69** |
| **OASIS-2, Mask x4** | | | | | | | | |
| PSNR ↑ | 23.77 | 15.13 | 23.23 | 26.72 | 26.74 | 26.77 | 26.86 | **27.06** |
| SSIM ↑ | 0.43 | 0.78 | 0.52 | 0.84 | 0.84 | 0.84 | 0.84 | **0.85** |

tizing DPS branch. If it does, we replace $\boldsymbol{thres}_{small}$ with $\boldsymbol{thres}_{big}$ ($\boldsymbol{thres}_{small} < \boldsymbol{thres}_{big}$) (Algorithm 3 line 2), which reduces the dependency to DPS guidance.

## 3. Results

We compare RefineGan (Quan et al., 2018), FDB (Mirza et al., 2023), DPS (Chung et al., 2022), I2SB (Liu et al., 2023), and CDDB (Chung et al., 2023). Table 1 shows the results for IXI and OASIS-2 datasets. Our proposed approach demonstrates superior performance compared to the previous sampling approaches with PSNR= 22.72 dB and SSIM= 0.68, with improvement of 0.45 dB for PSNR and 0.02 for SSIM. For OASIS-2 dataset, our sampling algorithm also demonstrates better results with PSNR= 27.06 dB and SSIM= 0.85, with improvement of 0.32 dB for PSNR and 0.01 for SSIM.

## 4. Conclusion

We introduced a new approach for MRI reconstruction and proposed a direct diffusion bridge sampling algorithm. The key component is utilizing both $k$-space and image domain information during the reconstruction process to balance perturbations caused by the DPS algorithm over DDB sampling. Our experimental results showed that our method outperforms other sampling algorithms on IXI and OASIS-2 datasets by exploiting information from two spaces. To the best of our knowledge, we are the first to analyze the efficiency of standard diffusion-based techniques application for DDB.

## Acknowledgments

This work was supported by the National Research Foundation of Korea (NRF) grant funded by the Korea government (MSIT) (No.2022R1C1C1008496), Institute of Information & Communications Technology Planning & Evaluation (IITP) grant funded by the Korea government (MSIT) (No.RS-2020-II201336, Artificial Intelligence Graduate School Program (UNIST), No.2022-0-00959, (Part 2) Few-Shot Learning of Causal Inference in Vision and Language for Decision Making).

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
