# OpenReview forum: "Consistent Direct Diffusion Bridge with Injection for MRI Reconstruction"
_MIDL.io/2024/Short_Papers — MIDL 2024 Short Papers_

### Official Review · Reviewer_WKdV · 2024-04-17

**Confidence:** 4
**Final Rating:** 5

**Review:**

This paper presents a compelling investigation into the use of diffusion-based image priors for reconstructing MRI images from undersampled k-space data. As one of the first works to explore this concept, it holds significant importance for the community and is poised to spark engaging discussions during the meeting.

---

### Decision · Program_Chairs · 2024-04-26

Accept